# Robotic Fast Patch Clamp in Brain Slices Based on Stepwise Micropipette Navigation and Gigaseal Formation Control

**DOI:** 10.3390/s25041128

**Published:** 2025-02-13

**Authors:** Jinyu Qiu, Qili Zhao, Ruimin Li, Yuzhu Liu, Biting Ma, Xin Zhao

**Affiliations:** 1Institute of Robotics and Automatic Information System, Tianjin Key Laboratory of Intelligent Robotics, Nankai University, Tianjin 300350, China; qiujinyu@mail.nankai.edu.cn (J.Q.); zhaoqili@nankai.edu.cn (Q.Z.); lrumin@mail.nankai.edu.cn (R.L.); liuyuzhu@mail.nankai.edu.cn (Y.L.); mabiting@mail.nankai.edu.cn (B.M.); 2Institute of Intelligence Technology and Robotic Systems, Shenzhen Research Institute of Nankai University, Shenzhen 518083, China; 3Beijing Advanced Innovation Center for Intelligent Robots and Systems, Beijing Institute of Technology, Beijing 100081, China

**Keywords:** robotic patch clamp, microoperation system, fuzzy logic control

## Abstract

The patch clamp technique has become the gold standard for neuron electrophysiology research in brain science. Brain slices have been widely utilized as the targets of the patch clamp technique due to their higher optical transparency compared to a live brain and their intercellular connectivity in comparison to cultured single neurons. However, the narrow working space, small scope, and depth of the field of view make the positioning of the operation’s micropipette to the target neuron a time-consuming task reliant on a high level of experience, significantly slowing down operation of the patch clamp technique in brain slices. Further, the current poor controllability in gigaseal formation, which is the key to electrophysiology signal recording, significantly lowers the patch clamp success rate. In this paper, a stepwise navigation of the micropipette is conducted to accelerate the positioning process of the micropipette tip to the target neuron in the brain slice. Then, a fuzzy proportional–integral–derivative controller is designed to control the gigaseal formation process along a designed resistance curve. The experimental results demonstrate an almost doubled patch clamp technique speed, with a 25% improvement in the success rate compared to the conventional manual method. The above advantages may promote the application of our method in brain science research based on brain slice platforms.

## 1. Introduction

The patch clamp technique has been recognized as the gold standard for the electrophysiological characterization of individual neurons in brain research [1]. At present, brain slices have been widely applied as the operation target of the patch clamp technique in brain science research [2,3,4]. In comparison to the animal brain environment, brain slices cut to only 200–400 μm in thickness have better optical transparency, facilitating observation of the target neurons in the patch clamp technique under microscopy [5]. Further, in comparison to single cultured neurons, part of the electrical connectivities between the neurons in brain slices can be saved due to the thickness of the brain slices [6]. Due to the above advantages, the brain slice has become an ideal platform for brain science research based on the patch clamp technique.

The patch clamp technique in brain slices is shown in Figure 1. The operator usually positions a micron-sized micropipette inserted with a silver line to approach the target neuron, aspirate part of the neuron membrane into the micropipette to form a giga Ω-scale seal (gigaseal) to shield from environmental noise, break the aspirated neuron membrane (break-in), and finally measure the electrophysiological signals of the target neuron [7].

However, the current patch clamp technique in brain slices still faces some challenges. To facilitate the observation of the neurons buried in the brain tissue environment, the current patch clamp technique in brain slices usually needs to be performed under a high-objective lens. The narrow working space, small scope, and depth of the field of view (FOV) under a high-objective lens make the navigation of the operation’s micropipette tip to the target neuron a challenging task. As shown in Figure 1, the operator first needs to move the micropipette tip with a random initial position to the relatively small field of view and then lowers it down to the target neuron surface after repeated back-and-forth focus of the micropipette tip and neuron, which is time-consuming (80% of the total time before recording) and requires a high level of experience [8].

Furthermore, to detect extremely weak cellular electrophysiological signals, which are usually at the picoampere level (10^−12^ A), a giga Ω-level seal needs to be formed between the inner wall of the measurement micropipette and the neuron surface to shield from environmental electrical disturbances [9]. The operator usually uses a mouth pipette or a syringe (see Figure 1) to exert constant aspiration pressure inside the micropipette to facilitate the formation of the gigaseal [3,7]. As the aspiration pressure is mainly determined by the operator’s experience, the gigaseal formation results are highly random, easily leading to gigaseal formation and patch clamp failures. In summary, the above two issues significantly limit the operation speed of the patch clamp technique and lower its success rate, which finally limits the broader and more advanced application of the patch clamp technique in brain slice platforms. Therefore, a novel micropipette navigation process and gigaseal formation method are highly desired to improve the efficiency of patch clamp operation.

Automated planar patch clamp systems integrate an array of microholes in a planar chip connecting to a pump to aspirate and operate multiple suspended cells at the same time [10]. However, these systems are only applicable for suspended cells but are not suitable for the patch clamp technique for adherent cells or brain slice neurons. Automated blind patch clamp systems use a micropipette to enter the brain slices or a live animal brain and then finish the gigaseal formation, break-in, and signal recording on the encountered neuron based on the detected resistance without visual feedback [11,12]. However, because of the lack of visual guidance, encountering the target neuron is rather random, which significantly lowers the success rate of the patch clamp technique. As an improvement, differential interference contrast (DIC) [8] and two-photon microscopy [13] have been equipped to visualize the target neurons in brain slices and in vivo, respectively, with the patch clamp technique. However, since the micropipette navigation process to the target neuron and the gigaseal formation control method still mimic manual operation, patch clamp technique speed and success rate have no significant improvements compared to the manual method. In previous research, we have proposed a robotic patch clamp method based on 3D cell morphology [14]. The success rate of the patch clamp technique was improved through the selection of an appropriate contact point between the micropipette and the neuron surface. However, the involved online 3D cell morphology operation procedure slows down the entire patch clamp operation process. In summary, a robotic patch clamp operation method with a new micropipette navigation method and a new gigaseal formation method is still desired to improve the operation speed and success rate of the patch clamp technique.

In this paper, a robotic stepwise micropipette navigation process and a gigaseal formation control method for the patch clamp in brain slices are proposed. First, a three-step navigation of the micropipette to the target neuron is designed, including coarse search and localization of the micropipette under the large field of view of a low-objective lens, fine localization of the micropipette tip and its lowering to approach the neuron under the small field of view of a high-objective lens, and finally, precise landing on the target neuron surface based on the resistance model. The stepwise micropipette navigation process was proposed to accelerate the positioning process of the micropipette from its initial position to target the neuron surface. Then, a force analysis of the gigaseal formation process was performed to design a suitable resistance curve to facilitate the gigaseal formation process. Further, a fuzzy proportional–integral–derivative (FPIDC) controller is designed to control seal resistance along the designed resistance curve to conduct the gigaseal formation process. Finally, a robotic fast patch clamp operation process was established to accelerate the patch clamp’s speed and improve its success rate. The experimental results of the whole-cell patch clamp on the pyramidal neurons in the visual cortex demonstrate that the proposed system almost doubled the patch clamp technique’s speed before signal recording, with a 25% improvement in the success rate compared to the conventional manual method. Further, normal action potentials were recorded from target neurons, demonstrating that our system does not have negative influences on cell electrophysiological activities. The above advantages may promote the application of our method in brain science research based on the brain slice platform.

## 2. System Setup

As shown in Figure 2, an immovable stage mounted on a vibration-isolation table is utilized to position the slice chamber containing the brain slices. A slice anchor is used to stabilize the slice in the slice chamber. A standard upright microscope (Eclipse FN1, Nikon, Tokyo, Japan) mounted on the motor stage capable of moving in the *X*-*Y* plane (with a travel range of 50 mm × 50 mm, a maximum speed of 1 mm/s, and repeatability of ±0.1 μm; MP285, Sutter Instrument, Novato, CA, USA) is utilized to observe the neurons. A CCD camera (IR-2000, DAGE-MTI, Michigan City, IN, USA) is mounted on the microscope to acquire images at 60 fps. A motorized focus device (with repeatability of ±0.1 μm; ES10ZE, Prior, Cambridge, UK) is installed on the microscope to position the focal plane. An *X*-*Y*-*Z* micromanipulator (with a working space of 50 mm × 50 mm × 50 mm; MP285, Sutter Instrument, Novato, CA, USA) is used to position the electrode micropipette. A signal amplifier (Multiclamp 700B, Molecular Devices, San Jose, CA, USA) and a data acquisition device (DAQ USB-6211, National Instruments, Austin, TX, USA) are used for voltage control and signal acquisition. An in-house-developed pneumatic control box provides an aspiration pressure with a range from -5 psi to 15 psi and a resolution of 10 Pa. The custom-developed human–machine interface (HMI), written in C++, controls all the aforementioned hardware in a multi-threaded manner and implements a state machine and event-driven architecture, referencing [15]. The HMI allows the operator to monitor the system state in real time, such as resistance values, pressure values, manipulator positions, imaging, error reports, etc. It also supports manual intervention during automated patch clamp processes.

The 4–6-week-old female mice (C57BL/6N) were prepared for robotic patch clamp experiments with neurons in mouse brain slices. After the blocks of tissue were removed through surgical procedures, they were immediately immersed in ice-cold artificial cerebrospinal fluid (ACSF): (in mm) 125 NaCl, 1.25 NaH_2_PO_4_, 2.5 KCl, 6.2 D-Glucose, 25 NaHCO_3_, 1 MgSO_4_, 2 CaCl_2_ (pH = 7.3, osmolarity ∼295 mmol/kg), saturated with 95% O_2_ and 5% CO_2_. The slices were first cut at a thickness of 300 mm with a vibrating blade microtome (700smz, Campden, Leics, UK) and then incubated at 37 °C for 30 min in ACSF.

The micropipette was fabricated from filamented capillaries (1.5 mm outer diameter, 0.86 mm inner diameter; BF150-86-10, Sutter Instrument). It was pulled by a pipette puller (P97, Sutter Instrument), followed by tip polishing with a microforge (MF-900, Narishige, Tokyo, Japan) to smooth the opening. Before being installed on the micromanipulator, the micropipette was filled with an electrolytic solution: (in mm) 120 K-gluconate, 4 KCl, 10 HEPES, 0.3 Na_2_GTP, 4 MgATP, 10 phosphocreatine and 0.5% biocytin (pH = 7.2, osmolarity ~300 mmol/kg). When loading the micropipette into the holder, the 200 μm diameter silver-chloride-coated electrode wire connected to the amplifier must be immersed in the electrolyte solution inside the micropipette. If the bath resistance of the micropipette is between 4 and 7 MΩ, the micropipette tip is considered to have an appropriate shape.

## 3. Key Methodologies and Technologies

### 3.1. Stepwise Micropipette Navigation

After incubation, the brain slice is laid out in the slice chamber and stabilized with a slice anchor. The FOV under the 4× objective lens is then manually moved to the target brain region according to the brain slice atlas. The micropipette, filled with the electrolytic solution, is then loaded into the micropipette holder by the operator. A low positive pressure of 0.5 psi is applied inside the micropipette, which generates a fine flow out of its opening to reduce clogging issues when the tip is immersed in the bath solution. After the above preparatory work, the designed robotic stepwise micropipette navigation process begins, including the following three steps: coarse navigation to search and localize the micropipette and brain slice under the FOV of the 4× objective lens, fine navigation to search and localize the micropipette tip and lower it down to approach the target neuron under the FOV of the 40× objective lens, and precise navigation to land on the target neuron surface based on the resistance model.

#### 3.1.1. Coarse Navigation

As shown in Figure 3a, after the preparatory work is completed, the target brain region on the brain slice is focused within the FOV under the 4× objective lens. The height of the brain slice surface is recorded as $H_{s}$, which is obtained from the height of the current focal plane.

As shown in Figure 3b, the focal plane is then raised by $H_{l}$ from the brain slice to provide enough micropipette movement workspace and a clean imaging background for subsequent image processing. The micropipette tip is moved into the FOV by sweeping in a zigzag pattern [16]. The micropipette is then adjusted to be focused using the normalized variance method [17]. The focus measure, *M*, changes as the micropipette is adjusted. When *M* reaches the global maximum, the micropipette is considered in focus.(1)$$\begin{array}{c} M=\frac{1}{H\cdot W\cdot \mu }\sum_{Hight}{\sum_{Width}\left(I(x,y)-\mu \right)}^{2} \end{array}$$
where μ is the average image intensity of the image, *H* and *W* are the height and the width of the image, respectively, and I(x,y) is the grey level intensity of pixel (x,y). Since the micropipette is mounted at a tilting angle with the tip side down, the in-focus region is detected using the quad-tree recursive algorithm [16]. The rightmost in-focus region, which is the micropipette tip, is then brought into focus by moving the micropipette upwards.

As shown in Figure 3c, after focusing, the height of both the focal plane and the micropipette tip is recorded as $H_{l}+H_{s}$. The micropipette tip is then tracked in real time within the image using corner detection based on the contour sharp degree [18] defined as(2)$$\begin{array}{c} S_{\mathrm{sharp}}=1-\frac{\left|P_{i-k}P_{i+k}\right|}{\left|P_{i}P_{i-k}\right|+\left|P_{i}P_{i+k}\right|} \end{array}$$
where $P_{i}$ represents the pixel coordinates of the *i*-th point on the micropipette contour, and $P_{i-k}$ and $P_{i+k}$ represent the pixel coordinates of the *k*-th point before and after the point $P_{i}$ on the contour, respectively. The Euclidean distance between $P_{i-k}$ and $P_{i+k}$ is denoted as $\left|P_{i-k}P_{i+k}\right|$. The point $P_{i}$ with the maximum $S_{\mathrm{sharp}}$ on the contour is considered the micropipette tip. A proportional–integral–derivative (PID) controller is then used to move the micropipette tip to the coordinates $\left(x_{40\times },y_{40\times }\right)$ of the 40× objective lens FOV, calibrated within the 4× objective lens FOV.

Then, as shown in Figure 3d, the micropipette and focal plane are lowered by $H_{l}-\Delta H_{s}$ and $H_{l}-\Delta H_{s}-\Delta H_{f}$, respectively, resulting in final positions at heights of $H_{s}+\Delta H_{s}$ and $H_{s}+\Delta H_{s}+\Delta H_{f}$, respectively. Considering the focusing errors resulting from the large depth of field (DOF) of the 4× objective lens, a safety height redundancy $\Delta H_{s}$ is set here to prevent the micropipette tip from touching the brain slice. Similarly, considering the height difference between the focal planes of the 4× and 40× objective lenses, the redundancy $\Delta H_{f}$ is set here to ensure that the focal plane of the 40× objective lens is positioned above the micropipette tip, facilitating subsequent fine navigation.

#### 3.1.2. Fine Navigation

As shown in Figure 3e, the objective lens is first switched to 40× magnification. After the coarse navigation, the micropipette tip is initially positioned directly below the FOV. As the focal plane is lowered at a speed of 5 μm/s, the focus on the inclined micropipette gradually “moves” from its root to its tip. During this process, a complete image of the micropipette is reconstructed using the motion history images (MHI) algorithm [19], and the rightmost point of the reconstructed micropipette is considered the tip coordinate (see the right part of Figure 3e. Based on this reconstruction, the micropipette tip is automatically focused and detected, and its coordinates in the image are recorded as $\left(x_{t},y_{t}\right)$.

As shown in Figure 3f, the focal plane is first lowered by $\Delta H_{\mathrm{surface}}$ and then descends simultaneously with the micropipette tip at a speed of 5 μm/s, maintaining a constant height difference between them to prevent the micropipette tip from touching the brain slice surface. The descent continues until the normalized variance method [17] identifies that the focal plane has reached the surface of the brain slice.

As shown in Figure 3g, a healthy pyramidal neuron on the surface of the brain slice is selected by the operator, and its coordinates $\left(x_{n},y_{n}\right)$ are determined in the image through mouse clicks. Based on the image coordinates of the micropipette tip $\left(x_{t},y_{t}\right)$ and the target neuron $\left(x_{n},y_{n}\right)$, as well as the calibrated angle θ between the image coordinate system $X_{i}-Y_{i}$ and the micromanipulator motion coordinate system $X_{m}-Y_{m}$, the micropipette tip is horizontally moved to the target neuron’s coordinates $\left(x_{n},y_{n}\right)$ according to(3)$$\begin{array}{c} \left[\begin{array}{c} \Delta x_{m}\\ \Delta y_{m} \end{array}\right]=D\left[\begin{array}{cc} \cos\theta & -\sin\theta \\ \sin\theta & \cos\theta \end{array}\right]\left[\begin{array}{c} x_{t}-x_{n}\\ y_{t}-y_{n} \end{array}\right] \end{array}$$
where *D* is the calibrated ratio between pixels and microns. At this point, the micropipette tip is directly above the target neuron, maintaining a height difference of $\Delta H_{\mathrm{surface}}$ from the surface of the brain slice.

#### 3.1.3. Precise Navigation

As shown in Figure 3h, after fine navigation, the micropipette tip is first lowered by $\Delta H_{\mathrm{surface}}$ to reach the surface of the brain slice. At this point, the tip is close to the target neuron located below the brain slice surface. However, in the complex background of brain slice images, it is challenging to accurately detect the vertical distance between the micropipette tip and the upper surface of the neuron using image guidance. To address this issue, we previously established an electrical model of the gap space between the micropipette tip and the neuron membrane [14]. As shown in the rightmost part of Figure 4, the gap space resistance, $R_{g}$, can be calculated according to(4)$$\begin{array}{c} R_{g}=\int_{R_{1}}^{R_{2}}\rho \frac{dr}{s}=\int_{R_{2}}^{R_{1}}\rho \frac{dr}{\pi r(a+b)}=\int_{R_{1}}^{R_{2}}\rho \frac{dr}{2\pi rd}=\frac{\rho }{2\pi d}\ln\frac{R_{2}}{R_{1}} \end{array}$$
where $R_{1}$ is the radius of the virtual superconductor, $R_{2}$ is the radius of the cylinder gap space, ρ is resistivity, *s* is the cross-section area, *a* and *b* are the highest and lowest height of the gap space, respectively, and *d* is the distance between the micropipette opening center and neuron surface along the micropipette axis direction. More details of this resistance model can be found in [14].

As the micropipette tip approaches the neuron surface, the decrease in *d* leads to an inverse increase in $R_{g}$ according to Equation (Equation 4), and this causes a rise in the measured bath resistance $R_{b}$ of the micropipette. In this process, when the micropipette resistance $R_{b}$ increases by a threshold of $0.1R_{b}$, the micropipette opening is considered to have made contact with the upper surface of the target neuron. Then, the constant positive pressure inside the micropipette is released, allowing the micropipette opening to be covered by the neuron’s membrane, thus facilitating subsequent membrane aspiration for gigaseal formation.

#### 3.1.4. Stepwise Navigation Process from Coarse to Fine to Precise

The relationship between the stepwise navigation process from coarse (ten-micron accuracy) to fine (micron accuracy) to precise (sub-micron accuracy) is illustrated in Figure 4. First, as shown in the green box in Figure 4, coarse navigation moves the micropipette to the target region (the orange box) within a 2 mm × 2 mm FOV under the 4× objective lens. Through this coarse visual guidance with ten-micron accuracy, the micropipette is brought into the small FOV (200 μm × 200 μm) under the 40× objective lens, with its vertical height relative to the brain slice determined. Then, as shown in the orange box in Figure 4, fine navigation moves the micropipette close to the target neuron in the *X*-*Y* plane under the 40× objective lens with micron accuracy using visual guidance.

Finally, during the process of pressing the micropipette onto the neuron, visual guidance relies on neuron deformation to confirm contact. However, by the time deformation is observed, the micropipette may have already pressed too deeply into the neuron, increasing the risk of damage. Therefore, as shown in the red box in Figure 4, precise navigation non-invasively measures the vertical height difference between the micropipette tip and the target neuron with sub-micron accuracy, enabling the micropipette tip to be gently pressed onto the neuron membrane.

### 3.2. Gigaseal Formation Control

Due to the interaction between the phospholipid bilayer and the glass surface, the membrane patch aspirated into the micropipette adheres tightly to its inner wall. This tight adhesion restricts the current flowing through the gap between the membrane and the micropipette, forming a seal with gigaohm-level resistance (gigaseal) [9]. The gigaseal effectively shields external noise, ensuring that the current measured by the micropipette electrode primarily originates from the ion channels on the neuron. Therefore, the success rate of gigaseal formation significantly influences the success rate of patch clamp recordings.

In conventional gigaseal formation, operators use a syringe or mouth suction to aspirate the neuron membrane into the micropipette (see Figure 1) until the micropipette resistance reaches the gigaohm level. Since gigaseal formation is a complex dynamic process, aspiration that is too fast or too slow may cause the aspirated membrane to fold [20], tear [21], or detach [22]. Therefore, conventional gigaseal formation heavily relies on the operator’s experience and limits the success rate of patch clamp recording. To improve the success rate of gigaseal formation, for the first time, this paper designs a control strategy for the gigaseal formation process.

First, the force exerted on the membrane aspirated into the micropipette tip during gigaseal formation is analyzed to design an appropriate gigaseal formation process. According to the literature [23], the membrane is primarily influenced by the aspiration force $F_{p}$, membrane–glass adhesion force $F_{a}$, cellular viscoelasticity $F_{v}$, and cytoskeletal forces $F_{c}$, as shown in Figure 5. The force pulling the membrane is the aspiration force, while the opposing forces mainly arise from cellular viscoelasticity, cytoskeletal forces, and membrane–glass adhesion. Based on the aforementioned force analysis, in the initial stage of gigaseal formation, only a small portion of the membrane enters the micropipette opening, resulting in a minimal membrane–glass adhesion force. To prevent membrane detachment from the micropipette, the membrane needs to be rapidly aspirated. Assuming that the resistance along the adhesion area per unit length is uniformly distributed [24], the resistance needs to increase at an accelerated rate during this stage. In the middle stage of gigaseal formation, the aspiration pressure needs to be adjusted to balance the adhesive force and viscoelastic force, ensuring that the adhered membrane creeps smoothly along the inner wall of the micropipette. During this stage, the resistance should increase steadily at a constant rate as the adhesion area expands smoothly. In the final stage, the membrane inside the micropipette has reached sufficient length, and aspiration should be stopped to prevent overstretching and potential damage to the membrane. After aspiration stops, the resistance slowly increases due to the further adhesion of the membrane to the inner wall and eventually stabilizes. Based on the three stages described above, the desired resistance trajectory for the gigaseal formation process is designed in this paper as follows: initially, the resistance increases with an acceleration $a_{R}$, then increases with a constant rate $v_{R}$. When it reaches a sufficiently high resistance $R_{t}$, the aspiration stops, allowing the seal resistance to stabilize. When the absolute value of the resistance change rate remains below 10 MΩ per second, the resistance is considered stable and considered as the final seal resistance.

Considering that gigaseal formation is a complex dynamic process that is highly uncertain and nonlinear, and there are significant variations in the experimental conditions across individual trials, a fuzzy PID controller (FPIDC) is designed to control the gigaseal formation process along the desired resistance trajectory. The controller with self-tuning parameters adjusted by fuzzy logic allows the integration of the experience and knowledge of skilled operators into the controller design [25]. Here, the control law of the applied pressure u(t) in the micropipette is(5)$$\begin{array}{c} u\left(t\right)=k_{p}\left(t\right)e\left(t\right)+k_{i}\left(t\right)\int_{0}^{t}e(\tau )d\tau +k_{d}\left(t\right)\dot{e}\left(t\right) \end{array}$$
where e(t) is the tracking error of the resistance trajectory, and the self-tuning parameters are updated by(6)$$\begin{array}{c} \left\{\begin{array}{c} k_{p}\left(t\right)=k_{p0}+\Delta k_{p}\left(e\left(t\right),\dot{e}\left(t\right)\right)\\ k_{i}\left(t\right)=k_{i0}+\Delta k_{i}\left(e\left(t\right),\dot{e}\left(t\right)\right)\\ k_{d}\left(t\right)=k_{d0}+\Delta k_{d}\left(e\left(t\right),\dot{e}\left(t\right)\right) \end{array}\right. \end{array}$$
with the initial control gains $k_{p0}$, $k_{i0}$, $k_{d0}$ and the time varying incremental gains $\Delta k_{p}$, $\Delta k_{i}$, $\Delta k_{d}$.

The fuzzy linguistic variables are defined as LN (large negative), MN (medium negative), S (small), MP (medium positive), LP (large positive). The IF–THEN fuzzy rules for gains $\Delta k_{p}$, $\Delta k_{i}$, and $\Delta k_{d}$ are listed in Table 1 (e.g., IF *e* is LN and $\dot{e}$ is LN, THEN $\Delta k_{p}$ is LP, $\Delta k_{i}$ is LN, and $\Delta k_{d}$ is S). Fixed PID parameters $k_{p0}$, $k_{i0}$, $k_{d0}$ and triangular membership functions (trimf) used for input and output variables are shown in Table 2. The subsequent defuzzification step can be achieved by the use of the center-of-area method [26].

## 4. Experiments

A total of 40 operations targeting healthy pyramidal neurons in the primary visual cortex, selected from 10 brain slices obtained from five 4-week-old mice, were divided into two groups. In the first group, four operators with one year of patch clamp experience each performed five manual whole-cell patch clamp operations. Generally, a conventional manual micropipette positioning operation is divided into the following three steps:Step 1: Moving the micropipette into the field of view;Step 2: Positioning the target neuron;Step 3: Pressing the micropipette tip onto the target neuron, which is indicated by the dimple observed on the neuron surface upon contact.

In the second group, 20 trials were operated using the proposed robotic patch clamp process. Different from conventional manual operation, the robotic stepwise micropipette navigation process, as described in Section 3.1, is divided into three steps: coarse navigation, fine navigation, and precise navigation. The entire robotic whole-cell patch clamp process is shown in Figure 6a. The control diagram of the system is presented in Figure 6b. The parameters set in the robotic process are listed in Table 3.

### 4.1. Comparison of Average Time

The time taken for each step of the micropipette navigation process in both groups was recorded to evaluate their performance. The average time of each step of successful manual operations and robotic processes is shown in Figure 7a. The results indicate that, for the micropipette navigation process, the average time required for the developed robotic process (157.1 ± 6.0 s, n = 18 successful trials) was 38.7% shorter than that for manual operation performed by operators with one year of patch clamp experience (256.3 ± 26.9 s, n = 18 successful trials), and the time for manual operation exhibited a large standard deviation across different operators and individual trials. In manual operation step 1, the operator needs to move the micropipette from an unknown position outside the FOV into the small FOV of the 40× objective lens. In contrast, in the coarse navigation of the proposed robotic process, the micropipette is moved into the FOV of the 4× objective lens, which has a larger FOV and DOF, in less time. In manual operation step 2, due to the unknown distance between the micropipette and the brain slice surface, the operator needs to lower both the micropipette and the focal plane together until the brain slice surface is located. In contrast, in the fine navigation of the proposed robotic process, based on the recorded brain slice height from the coarse navigation, the micropipette and focal plane can quickly approach the brain slice surface. In manual operation step 3, the operator needs to adjust the focal plane back and forth to estimate the vertical relative distance between the micropipette tip and the target neuron surface while slowly lowering the micropipette until a dimple is observed on the neuron surface caused by the micropipette tip pressing. In the precise navigation of the proposed robotic process, the vertical relative distance between the micropipette tip and the neuron surface is calculated using the resistance model, enabling the micropipette to press the neuron in less time.

### 4.2. Comparison of Success Rate and Final Gigaseal Resistance

The success rates of micropipette navigation, gigaseal formation, and membrane break-in for the two groups are shown in Figure 7b. For the micropipette navigation process, the manual operation achieved a success rate of 90% (18/20), with failures attributed to the operator not observing the dimple on the neuron surface after gently pressing the micropipette tip onto the target neuron. The robotic process achieved the same success rate of 90% (18/20), with failures resulting from the resistance not reaching the threshold during the precise navigation step. Failures in both groups occurred during the step of pressing the micropipette tip onto the neuron, potentially influenced by factors such as neuron morphology, elasticity, and the depth of the neuron within the brain slice.

For gigaseal formation, the manual operation and the robotic system achieved success rates of 66.7% (12/18) and 88.9% (16/18), respectively, with average final gigaseal resistances of 2.28 ± 0.62 GΩ (n = 12) and 2.77 ± 0.36 GΩ (n = 16) (*p* = 0.021) (see Figure 7c), and with the average root mean square error (RMSE) of tracking errors of 75.37 MΩ (n = 16) (see Figure 7d). For membrane break-in, the manual operation and the robotic process achieved success rates of 58.3% (7/12) and 75.0% (12/16), respectively.

The experimental results demonstrate that the robotic system (12 successful trials out of 20) shows a 25% improvement in the patch clamp success rate compared to the conventional manual method (7 successful trials out of 20). The results indicate that the proposed gigaseal formation method improved the success rates of gigaseal formation and membrane break-in, while achieving higher gigaseal resistances. This improvement could be attributed to the method facilitating tighter adhesion between the membrane and the micropipette inner wall, resulting in a higher-quality gigaseal structure. A more stable gigaseal structure may better withstand the disturbance of the pressure pulse during membrane break-in.

### 4.3. Electrophysiological Signal Recordings

The image of the micropipette and target neuron is shown in Figure 8a. After membrane break-in, the whole-cell configuration was established. Action potentials generated by one of the target neurons in response to a 1000 ms, 100 pA current pulse injected into the micropipette are shown in Figure 8b. Excitatory postsynaptic currents (EPSCs) recorded from one of the target neurons by voltage-clamping the micropipette at −70 mV are shown in Figure 8c. These results demonstrate that the system is capable of effectively recording the electrophysiological activity of neurons.

## 5. Discussion

The high magnification of objective lenses in micromanipulation often results in a narrow working space. Adding rotational degrees of freedom to the microscope stage has been reported to effectively expand working space [27]. However, different from traditional micromanipulation systems, the patch clamp electrophysiology system requires both the biological sample and the micromanipulator to be placed on a fixed platform rather than a movable microscope stage. This fixed placement ensures stable connections between the sample and the measurement electrode and reduces noise. As an optional approach, additional degrees of freedom could be added to the micromanipulator to expand the working space of the micropipette electrode and enable faster micropipette navigation.

The methods in this paper still have some limitations. Since the mechanism of gigaseal formation is not fully understood yet, and the neuron membrane inside the micropipette tip is difficult to observe under bright-field microscopy, a simple desired trajectory is used for gigaseal formation, and the model-free controller in this paper still has some tracking errors along the trajectory. As further research is conducted on gigaseal formation, more suitable desired trajectories and improved controllers are expected to enhance the gigaseal formation process.

Additionally, the success rate of the patch clamp is significantly influenced by the health of the neuron, which can be affected by various factors such as the age of the mice, the incubation solution, room temperature, and the duration of the experiment. Even with the same operational method, the success rates varied across different days. Therefore, the factors mentioned above were controlled to be relatively similar in this paper to evaluate the performance of different methods.

## 6. Conclusions

In this paper, a robotic fast patch clamp process for brain slices is proposed based on stepwise micropipette navigation and gigaseal formation control. Compared to conventional manual operation, the three-step micropipette navigation almost doubles the speed of the micropipette–neuron contact process. Additionally, the designed FPIDC, which controls seal resistance along the desired trajectory, improves the gigaseal formation success rate by 25%.

The robotic patch clamp process is expected to significantly improve the throughput of electrophysiological recordings, thereby facilitating research where large amounts of electrophysiological data are needed, such as in neurodegenerative diseases and drug screening. Furthermore, the robotic one-micropipette navigation process could be easily adapted for robotic multi-micropipette patch clamp systems. This adaptation enables simultaneous electrophysiological recordings from multiple neurons, which is expected to facilitate research on neural circuits and the mechanisms of the nervous system.

## Figures and Tables

**Figure 1 sensors-25-01128-f001:**
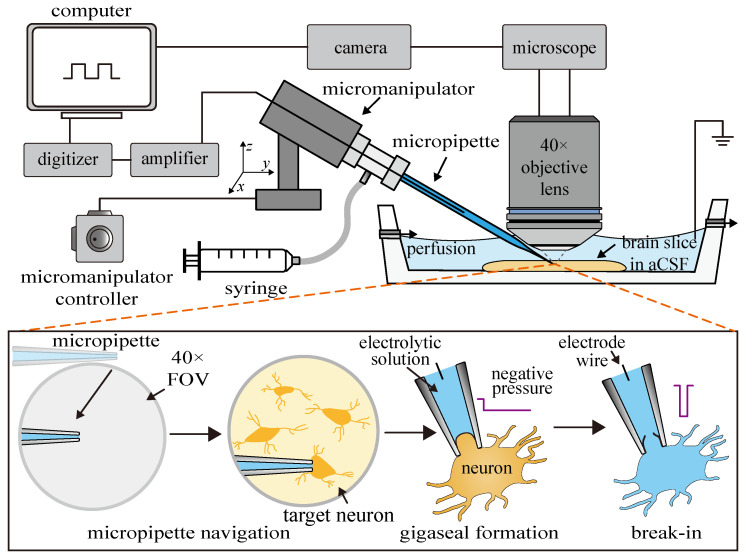
Schematic of conventional patch clamp system and its manual operation process.

**Figure 2 sensors-25-01128-f002:**
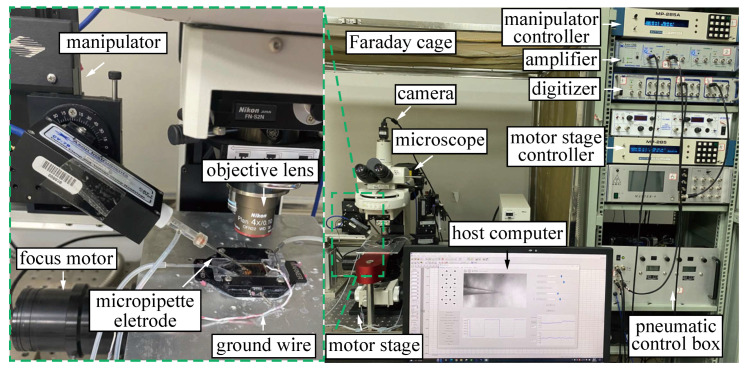
Robotic patch clamp system setup.

**Figure 3 sensors-25-01128-f003:**
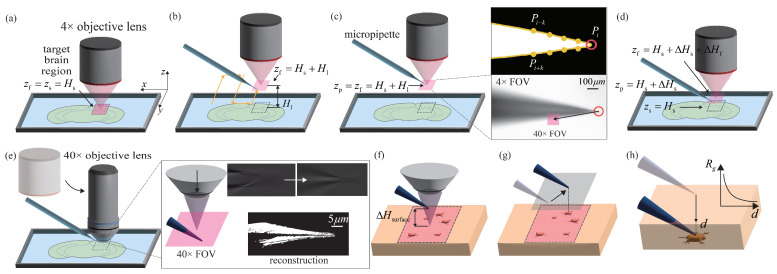
The designed robotic stepwise micropipette navigation process. (**a**) The state after completing preparatory work. (**b**) The focal plane is raised to load the micropipette, and the micropipette tip is then moved into the FOV by sweeping in a zigzag pattern and adjusted to be focused. (**c**) The micropipette tip is detected and moved to the pre-calibrated coordinates of the 40× objective lens FOV within the 4× objective lens FOV. (**d**) The micropipette and focal plane are lowered, immersing them in the bath solution. ((**a**–**d**) illustrate the coarse navigation process). (**e**) The objective lens is switched from 4× to 40×, and the micropipette tip is reconstructed and fine-positioned. (**f**) The micropipette maintains a constant height difference from the focal plane and descends together until the brain slice surface is focused. (**g**) The micropipette is moved horizontally to position directly above the target neuron. ((**e**–**g**) illustrate the fine-navigation process). (**h**) The distance between the micropipette tip and the target neuron is precisely positioned to gently press the micropipette tip onto the target neuron. ((**h**) illustrates the precise navigation process).

**Figure 4 sensors-25-01128-f004:**
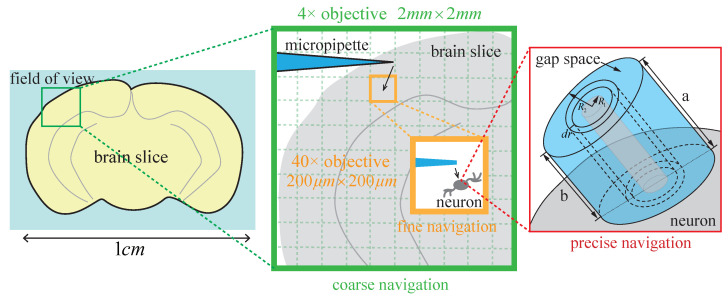
Stepwise navigation process from coarse to fine to precise.

**Figure 5 sensors-25-01128-f005:**
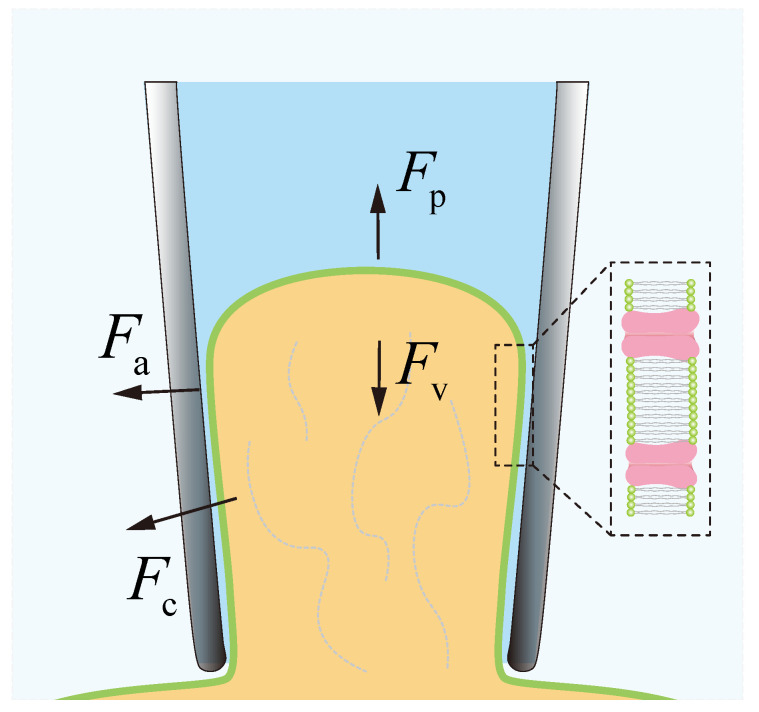
Schematic diagram of forces during gigaseal formation.

**Figure 6 sensors-25-01128-f006:**
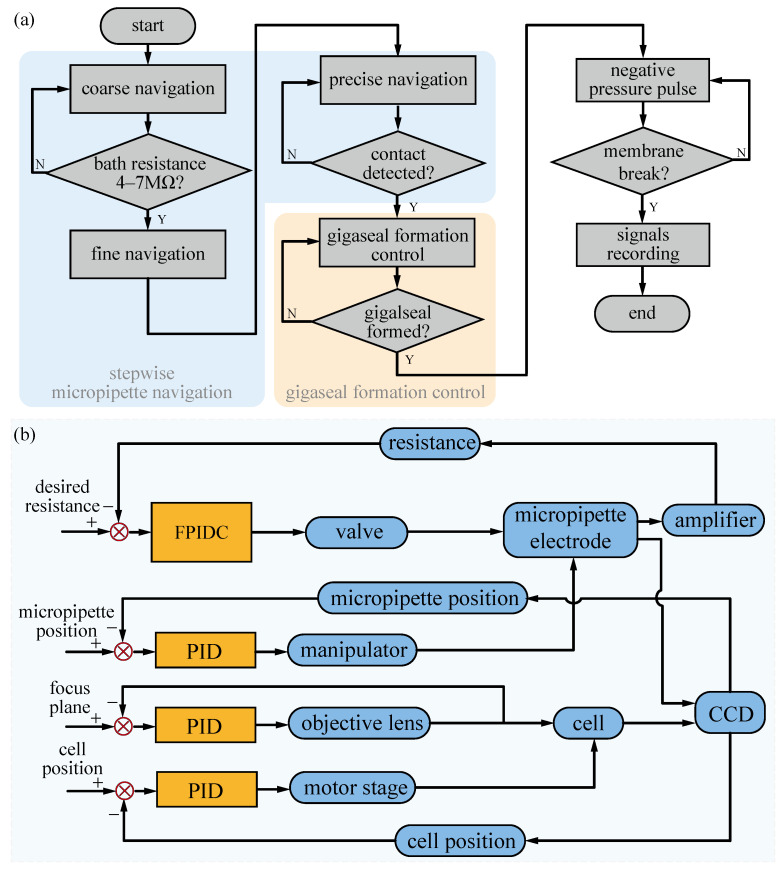
The robotic whole-cell patch clamp in brain slices. (**a**) The flow chart of the robotic whole-cell patch clamp process. (**b**) The control diagram of the robotic patch clamp system.

**Figure 7 sensors-25-01128-f007:**
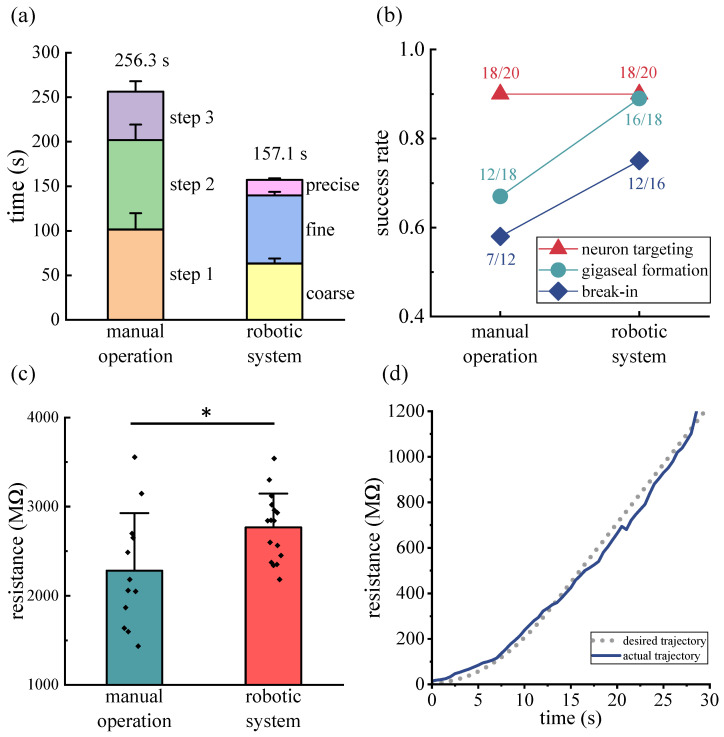
Experimental results. (**a**) Average time per step and total time for the micropipette navigation process with manual operation and the robotic system. (**b**) Success rates of micropipette navigation, gigaseal formation, and break-in with manual operation and the robotic system. (**c**) Final seal resistance obtained with manual operation and the robotic system (manual operation, 2.28 ± 0.62 GΩ (n = 12); robotic system, 2.77 ± 0.36 GΩ (n = 16), * *p* = 0.021). (**d**) Resistance tracking trajectory during one trial with the robotic system.

**Figure 8 sensors-25-01128-f008:**
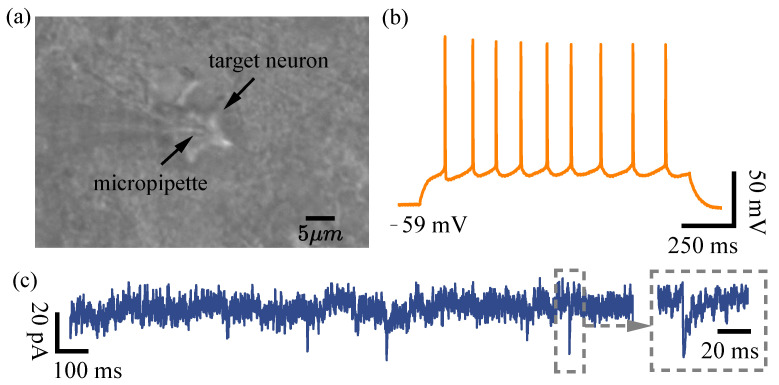
Signal recordings of the robotic whole-cell patch clamp process. (**a**) The micropipette and the target neuron during the recording. (**b**) Action potentials recorded from the target neuron with a 1000 ms-long current injection pulse at 100 pA. (**c**) EPSCs recordings with a holding potential of −70 mV.

**Table 1 sensors-25-01128-t001:** Fuzzy rules for gains $\Delta k_{p}$, $\Delta k_{i}$, and $\Delta k_{d}$.

	$\dot{e}$	LN	MN	S	MP	LP
*e*						
LN		LP/LN/S	LP/LN/S	MP/MN/LN	S/S/MN	S/S/S
MN		LP/LN/S	LP/LN/S	S/S/MN	S/S/S	S/S/S
S		LP/MN/S	MP/MN/S	S/S/S	MN/MP/S	MN/MP/S
MP		S/S/S	S/S/S	MN/S/S	MN/LP/LP	LN/LP/LP
LP		S/S/S	S/S/MP	MN/MP/MP	LN/LP/S	LN/LP/LP

**Table 2 sensors-25-01128-t002:** Fixed PID parameters $k_{p0}$, $k_{i0}$, $k_{d0}$ and triangular membership functions (trimf) consisting of five components (LN, MN, S, MP, and LP) for input variables *e* and $\dot{e}$ and output variables $\Delta k_{p}$, $\Delta k_{i}$, and $\Delta k_{d}$.

$k_{p0}$	$k_{i0}$	$k_{d0}$	*e*	$\dot{e}$	$\Delta k_{p}$	$\Delta k_{p}$	$\Delta k_{d}$
10	1	2	trimf [−20, 20]	trimf [−4, 4]	trimf [−2, 2]	trimf [−0.2, 0.2]	trimf [−0.4, 0.4]

**Table 3 sensors-25-01128-t003:** Parameters of the system.

Symbol	Value	Symbol	Value
$H_{l}$	4000 μm	*D*	0.12 μm/pixel
$\Delta H_{s}$	200 μm	$a_{R}$	4 MΩ/$s^{2}$
$\Delta H_{f}$	100 μm	$v_{R}$	52 MΩ/s
$\Delta H_{\mathrm{surface}}$	50 μm	$R_{t}$	1200 MΩ
θ	6.2°		

## Data Availability

The raw data supporting the conclusions of this article will be made available by the authors without undue reservation.

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
