# Peer review of "Robotic Fast Patch Clamp in Brain Slices Based on Stepwise Micropipette Navigation and Gigaseal Formation Control"

_sensors, 2025, doi:10.3390/s25041128_

Round 1
Reviewer 1 Report
Comments and Suggestions for Authors
Dear authors,
The manuscript presents relevant findings and is overall an excellent piece of work, with well-structured presentation. However, I recommend revising the structure by renaming the section currently titled Discussion to Conclusions, as this part primarily summarizes key results rather than engaging in an in-depth discussion.
Additionally, the noted Discussions section (more appropriately Conclusions) could benefit from conciseness, as certain sections contain redundant phrasing that could be streamlined for clarity and readability (For example, parts describing what has been accomplished with the advantages and limitations of your study and what should be pursued in the future should be presented in distinct paragraphs, ensuring a clear separation between them without interruptions or transitions between the two topics). A more direct presentation of findings and their implications would enhance the overall impact of the manuscript. May I also suggest expanding the implications for further research and how these findings contribute to the broader field with identifying specific directions for future studies.
Other than these minor revisions, the manuscript is well-written. Good luck!
Reviewer
Author Response
Comment:
Dear authors,
The manuscript presents relevant findings and is overall an excellent piece of work, with well-structured presentation. However, I recommend revising the structure by renaming the section currently titled Discussion to Conclusions, as this part primarily summarizes key results rather than engaging in an in-depth discussion.
Additionally, the noted Discussions section (more appropriately Conclusions) could benefit from conciseness, as certain sections contain redundant phrasing that could be streamlined for clarity and readability (For example, parts describing what has been accomplished with the advantages and limitations of your study and what should be pursued in the future should be presented in distinct paragraphs, ensuring a clear separation between them without interruptions or transitions between the two topics). A more direct presentation of findings and their implications would enhance the overall impact of the manuscript. May I also suggest expanding the implications for further research and how these findings contribute to the broader field with identifying specific directions for future studies.
Other than these minor revisions, the manuscript is well-written. Good luck!
Reviewer
Reply: Thank you very much for your valuable suggestions. In the revised manuscript, we have renamed the "Discussion" section to "Conclusion" and modified its structure and phrasing. This section now summarizes the methods and results of the paper, and also includes future research directions and potential broader applications of our work. Additionally, we have added a "Discussion" section, where we provide a discussion of some of the details and limitations of our work.
The revised conclusion is as follows: In this paper, a robotic fast patch clamp process for brain slices is proposed base on stepwise micropipette navigation and gigaseal formation control. Compared to conventional manual operation, the three-step micropipette navigation almost doubles the speed of the micropipette-neuron contact process. Besides, the designed FPIDC, which controls seal resistance along the desired trajectory, improves the gigaseal formation success rate by 25%.
The robotic patch clamp process is expected to significantly improve the throughput of electrophysiological recordings, thereby facilitating research where large amounts of electrophysiological data are needed, such as in neurodegenerative diseases and drug screening. Furthermore, the robotic one-micropipette navigation process could be easily adapted for robotic multi-micropipette patch clamp systems. This adaptation enables simultaneous electrophysiological recordings from multiple neurons, which is expected to facilitate research on neural circuits and the mechanisms of the nervous system.
Reviewer 2 Report
Comments and Suggestions for Authors
A single-channel patch clamp robotic system using only one micropipette is presented in this paper. Overall, this paper presents an interesting research topic. The following issues should be addressed before recommending this manuscript for publication.
1, What are the differences between fine navigation and precise navigation in terms of methods, principle, process and experimental results? The reviewer suggests authors including an explanation in Section 3 as a opening paragraph.
2, As authors said in page 2, “The narrow working space, small scope…. make the navigation of operation micropipette tip to the target neuron a challenging task”. Another optional method is to enhance the working space of the micro positioning stage. For more detail information authors can refer following paper “Design and analysis of a 3-DOF planar micromanipulation stage with large rotational displacement for micromanipulation system”.
3, How to control the vacuum pressure of the micropipette for grasping the neuro cells?
4, Why do authors choose fuzzy control methods for the system? What’s the meaning of “LN, MN, S, MP and LP”? Authors should explain it.
5, How to realize focusing the cell? Is it realized by the manual method or automatic method?
6, For a precise positioning/ focusing on a cell. The reviewer thinks that authors should label the cell. So my question is how to label the cell?
Author Response
A single-channel patch clamp robotic system using only one micropipette is presented in this paper. Overall, this paper presents an interesting research topic. The following issues should be addressed before recommending this manuscript for publication.
Comment 1: What are the differences between fine navigation and precise navigation in terms of methods, principle, process and experimental results? The reviewer suggests authors including an explanation in Section 3 as a opening paragraph.
Reply 1: Thank you so much for your valuable question and suggestion. We have added a paragraph in Section 3.1.4 and added Figure 4 to explain the stepwise relationship between fine and precise navigation in more detail.
Fine navigation aims to navigate the micropipette close to the neuron in the X-Y plane with micron accuracy using visual guidance. First, the motion history image (MHI) algorithm is used to reconstruct the shape of the inclined micropipette tip from left to right as the focal plane moves downward. The rightmost point of the reconstructed micropipette is to be the micropipette tip position. The image processing result is shown in the right part of Figure 3(e). Then, based on Equation (3), the X-Y coordinates of the micropipette tip and the target neuron are used to move the micropipette tip to the position directly above the target neuron.
Then, during the process of pressing the micropipette onto the neuron, visual guidance relies on the neuron deformation to confirm contact. However, by the time deformation is observed, the micropipette may have already pressed too deeply into the neuron, increasing the risk of damage. Therefore, precise navigation utilizes electrical guidance, enabling the non-invasive measurement of the relative distance between the micropipette and the neuron with sub-micron accuracy before contact occurs. This ensures that the micropipette tip is gently pressed onto the neuron membrane. The resistance model (see Equation (4)) in electrical guidance was proposed in our previous work and has been validated through both electrical simulations and experiments. For more details and experimental results of the resistance model, please refer to [14] of the reference list of the manuscript.
Comment 2: As authors said in page 2, “The narrow working space, small scope…. make the navigation of operation micropipette tip to the target neuron a challenging task”. Another optional method is to enhance the working space of the micro positioning stage. For more detail information authors can refer following paper “Design and analysis of a 3-DOF planar micromanipulation stage with large rotational displacement for micromanipulation system”.
Reply 2: Thank you for your valuable suggestion. We have reviewed the recommended paper and discussed it in Section 5 of the revised manuscript. Indeed, adding rotational degrees of freedom to the stage can effectively expand the working space for many micromanipulation tasks, such as microinjection. However, in electrophysiological systems, both the biological sample and the micromanipulator need to be placed on a fixed platform, rather than the microscope stage, to ensure stable connections between the sample and the measurement electrode and reduce noise from vibrations. The field of view is adjusted by moving the microscope. Nevertheless, this insightful idea could be applied in the future to increase the degrees of freedom of the micromanipulator, thereby expanding the working space and enabling faster micropipette navigation.
Comment 3: How to control the vacuum pressure of the micropipette for grasping the neuro cells?
Reply 3: Thank you very much for your question. As shown in Figure 2, the micropipette is connected to a pneumatic box via a pressure channel. In the pneumatic box, negative pressure is generated by vacuum generators, regulated by electronic pressure regulators, and then applied to the pressure channel. The system adjusts the voltage of the electronic pressure regulators to control the negative pressure in the micropipette.
Comment 4: Why do authors choose fuzzy control methods for the system? What’s the meaning of “LN, MN, S, MP and LP”? Authors should explain it.
Reply 4: We sincerely appreciate your question and suggestion. First, the mechanism of gigaseal formation is highly complex and still not fully understood, making it difficult to establish a precise model. Therefore, a model-free control method is required. Furthermore, patch clamp is a highly experience-dependent technique. Fuzzy logic allows the integration of experienced operators’ knowledge, making the system more accessible to users with little prior experience. Based on the above considerations, we chose the fuzzy control method.
LN (large negative), MN (medium negative), S (small), MP (medium positive), and LP (large positive) are fuzzy linguistic variables. The input variables, tracking error and tracking error rate, are fuzzified using the input membership functions formed by these five components. Then, using the fuzzy rules in Table 1 (e.g., IF e is LN and dot e is LN, THEN delta kp is LP, delta ki is LN, and delta kd is S), and the output membership functions, the output variables are defuzzified to obtain the output values. We have included the above explanations in the paragraph below Equation 6.
Comment 5: How to realize focusing the cell? Is it realized by the manual method or automatic method?
Reply 5: Thank you for your question. In healthy brain slices, neurons on the surface (within 20 microns of a 300-micron-thick slice) can be observed directly when the brain slice surface is automatically focused. In the step described in Section 3.1.2, we only need to obtain the X-Y coordinates of the target neuron, so precise focusing of single neuron in the Z-direction is not required in this step. The focus can also be manually adjusted during this process. For the Z-coordinate of the neuron, the resistance model is used to obtain it in Section 3.1.3. While this paper mainly focuses on micropipette navigation, detailed methods for neuron detection and autofocusing in brain slices can be found in our previous works (“In Positioning and Tracking of Neurons in Label-free Tissue Slice for Automatic Patch Clamping”, 2021 IEEE International Conference on Robotics and Biomimetics (ROBIO), and “Neuron Contact Detection Based on Pipette Precise Positioning for Robotic Brain-Slice Patch Clamps”, Sensors.). In the former, convolutional neural networks were used for neuron positioning and tracking. In the latter, an object detection model was used to obtain a three-dimensional bounding frame of neuronal cells, with the maximum plane of the cell considered as the focused plane.
Comment 6: For a precise positioning/ focusing on a cell. The reviewer thinks that authors should label the cell. So my question is how to label the cell?
Reply 6: Thank you for your question. To provide a precise answer, we will address both aspects of “label”: fluorescent labeling and image processing labeling.
For fluorescent labeling, calcium indicators and fluorescent proteins are commonly used to label neurons in the brain. Calcium indicators (e.g., OGB-1 AM) can be ejected from a micropipette into the target brain region, labeling neurons based on calcium activity. Fluorescent proteins can be expressed by injecting viral vectors (e.g., pAAV-CAG-NLS-GFP) to label cell membranes or by using transgenic mice (e.g., GAD67-GFP transgenic mice).
For image processing labeling, in our previous work, images of healthy and unhealthy neurons were classified and labeled by experienced operators. A neural network trained with these labels is capable of detecting and positioning healthy neurons within the images. For detailed methods, please refer to the following paper: “In Positioning and Tracking of Neurons in Label-free Tissue Slice for Automatic Patch Clamping”, 2021 IEEE International Conference on Robotics and Biomimetics (ROBIO).
Reviewer 3 Report
Comments and Suggestions for Authors
The authors propose a robotic system to navigate a micropipette over brain slices and form gigaseals. The paper thoroughly discusses the experimental setup and the developed control method. The experiments show the improvement of successful attempts compared with manual operation. To summarize, I find the paper well-written and interesting, and I have just a few minor comments listed below:
1. P. 2, Fig. 2. The xyz coordinate system in this figure is left-handed.
2. P. 3, l. 64. Abbreviation DIC appears without explanations.
3. P. 6, Eq. (2). This equation includes unexplained notations: i, k, Pi, and so on. I recommend the authors explain these notations in more detail, as well as the equation itself.
4. P. 7, Eq. (4). I recommend the authors revise this equation: for example, the authors mention parameters a and b under the equation, but it does not include these parameters. If possible, I also recommend the authors present a figure corresponding to this equation.
5. P. 7, l. 220. There is a typo “Gigalseal” in the subsection title.
6. P. 10, Fig. 5b. Shouldn’t the plus and minus signs around the summation blocks be swapped? (I believe the negative signs should correspond to the feedback signals.)
7. P. 10, l. 297. What is n = 18 here? Do the authors mean successful attempts?
8. P. 12, l. 327. What is P* = 0.021?
Author Response
The authors propose a robotic system to navigate a micropipette over brain slices and form gigaseals. The paper thoroughly discusses the experimental setup and the developed control method. The experiments show the improvement of successful attempts compared with manual operation. To summarize, I find the paper well-written and interesting, and I have just a few minor comments listed below:
Comment 1: P. 2, Fig. 2. The xyz coordinate system in this figure is left-handed.
Reply 1: Thank you so much for your kind reminder. We have corrected the coordinate system in Fig. 2 in the revised manuscript.
Comment 2: P. 3, l. 64. Abbreviation DIC appears without explanations.
Reply 2: Thank you very much for your kind reminder. We have added the full name for DIC, "differential interference contrast (DIC)," in the revised manuscript.
Comment 3: P. 6, Eq. (2). This equation includes unexplained notations: i, k, Pi, and so on. I recommend the authors explain these notations in more detail, as well as the equation itself.
Reply 3: We sincerely appreciate your suggestion. We have added more detailed explanations of the notations and the equation below Eq. (2) in the revised manuscript.
Comment 4: P. 7, Eq. (4). I recommend the authors revise this equation: for example, the authors mention parameters a and b under the equation, but it does not include these parameters. If possible, I also recommend the authors present a figure corresponding to this equation.
Reply 4: Thank you so much for your valuable suggestion. We have added a schematic diagram of the resistance model in the rightmost part of Figure 4 and revised Eq. (4) to make it more detailed. More details of this model can also be found in our previous paper [14] of the reference list of the manuscript.
Comment 5: P. 7, l. 220. There is a typo “Gigalseal” in the subsection title.
Reply 5: Thank you for your kind reminder. We have corrected this typo in the revised manuscript.
Comment 6: P. 10, Fig. 5b. Shouldn’t the plus and minus signs around the summation blocks be swapped? (I believe the negative signs should correspond to the feedback signals.)
Reply 6: Thank you for your kind reminder. We have corrected the signs in this figure (now Figure 6(b)) in the revised manuscript.
Comment 7: P. 10, l. 297. What is n = 18 here? Do the authors mean successful attempts?
Reply 7: Thank you for your question. Yes, n = 18 is the number of successful attempts. The average time is calculated based on these successful attempts. In our experiments, there were two failed attempts in each group. Since these failed attempts did not complete the full procedure, their times were not included in the average time calculation. We have added the explanation in the revised manuscript.
Comment 8: P. 12, l. 327. What is P* = 0.021?
Reply 8: Thank you for your question. The value P = 0.021 is the p-value from our statistical analysis. As it is below the 0.05 significance threshold, it suggests that the final gigaseal resistances achieved using manual operations and robotic processes are statistically significant.
Round 2
Reviewer 2 Report
Comments and Suggestions for Authors
This paper can be considered for publication in current form.